# Emulsion Surimi Gel with Tunable Gel Properties and Improved Thermal Stability by Modulating Oil Types and Emulsification Degree

**DOI:** 10.3390/foods11020179

**Published:** 2022-01-11

**Authors:** Shichen Zhu, Xiaocao Chen, Jiani Zheng, Wenlong Fan, Yuting Ding, Xuxia Zhou

**Affiliations:** 1College of Food Science and Technology, Zhejiang University of Technology, Hangzhou 310014, China; zhusc@zjut.edu.cn (S.Z.); 18238758268@163.com (X.C.); dingyt@zjut.edu.cn (Y.D.); 2Key Laboratory of Marine Fishery Resources Exploitment & Utilization of Zhejiang Province, Hangzhou 310014, China; 3National R&D Branch Center for Pelagic Aquatic Products Processing (Hangzhou), Hangzhou 310014, China; 4Zhejiang Yufu Food Co., Ltd., Hangzhou 310014, China; 18738337259@163.com (J.Z.); w17757191983@163.com (W.F.); 5Collaborative Innovation Center of Seafood Deep Processing, Dalian Polytechnic University, Dalian 116034, China

**Keywords:** surimi gel, thermal stability, gel properties, oil types, emulsification degree

## Abstract

High resistance to heating treatments is a prerequisite for ready-to-eat (RTE) surimi products. In this study, emulsion-formulated surimi gels were prepared, and the effects of oil types and emulsification degrees on the thermal stability of surimi gel were investigated. The results showed the gel properties of surimi gels were modulated by oil types and emulsification degrees. In detail, the rising pre-emulsification ratio caused the increase of the emulsifying activity index (EAI) and decrease of emulsifying stability index (ESI) for both emulsions. The larger droplet sizes of perilla seed oil than soybean oil may be responsible for their emulsifying stability difference. The gel strength, water retention, dynamic modulus and texture properties of both kinds of surimi gels displayed a firstly increased and then decreased tendency with the rising pre-emulsification ratios. The peak values were obtained as perilla seed oil emulsion with emulsification ratio of 20% group (P1) and soybean oil emulsion with emulsification ratio of 40% group (S2), respectively. Anyway, all emulsion gels showed higher thermal stability than the control group regardless of oil types. Similar curves were also obtained for the changes of hydrogen bond, ionic bond and hydrophobic interactions. Overall, perilla seed oil emulsion with emulsification ratio of 20% (P1 group) contributed to the improved thermal stability of surimi gels.

## 1. Introduction

Surimi-based products have been highly popular due to their unique texture and rich nutrients [1]. Specially, the emerging ready-to-eat (RTE) surimi products are drawing more and more interest owing to their great convenience. The essential sterilization steps have to be conducted for RTE surimi products, which is usually achieved by high-temperature treatments above 100 °C. However, the thermal treatments would inevitably destroy the molecular interactions supporting gel structure, further decreasing the technological properties of surimi products [2]. It is thereafter very important to enhance the thermal stability of RTE surimi products.

To date, various methods have been explored to enhance the thermal stability of surimi products. For example, some non-muscle protein ingredients are introduced into surimi systems, serving as fillers to form relatively stable gel products supported by the hydrogen bonding interactions [3]. Gao et al. [4] found that the addition of soy protein benefits for the improved gel strength of the meat paste based on their inherent gelation properties. Besides, the addition of non-muscle protein can also change the gel properties of surimi products via the modulation of microstructure. Previous studies have observed that porcine plasma protein contributed to the formation of a firm and uniform network structure, which significantly enhanced the gel properties of the bighead carp surimi [5]. Although the additions of non-muscle proteins largely affect the gel strength, some other attributes such as flavor are hardly improved.

The additions of exogenous oils have been confirmed to improve flavor, and color of surimi products except for texture [6]. Previous studies have been focused on the effects of animal fat on functional properties of meat [7,8]. However, the cardiovascular risks from the intake of animal fat with high saturated fatty acids cause the increasing healthy concerns [9]. In contrast, it has been established that vegetable oils show great health benefits due to their rich unsaturated fatty acid composition. More importantly, the pre-emulsified oils display the comparable quality contributions to animal fat [10]. For example, Youssef et al. [11] reported that the addition of pre-emulsified lipids significantly improved the microstructure of surimi products. The similar results are also supported by the recent reports from Zhou’s group [6,12,13]. The interactions between pre-emulsified oils droplets and myofibrillar proteins are responsible for the formation of interfacial protein membrane (IPF), which further occupy the voids in the protein gel matrix network. The presence of droplet fillers or copolymers reduced porosity and improved homogeneity of microstructure, and thereafter enhanced thermal stability of gel meat products [14]. It needs to be noticed that the effects of pre-emulsified oils on meat gels are largely determined by oil types and emulsification degree.

Perilla seed oil is a kind of edible vegetable oil and rich in polyunsaturated fatty acids (α-linoleic acid), which shows numerous healthy benefits (decreasing blood-lipoids and cholesterol, inflammation inhibition and anti-aging effects). The soybean oil has been widely used for the emulsified meat products. In this work, a comparable study on effects of perilla seed oil and soybean oil with pre-emulsification treatments on thermal stability of surimi products was performed. The inherent properties of pre-emulsification particles and gel properties of emulsion surimi gels were characterized. Furthermore, the soluble protein content involving the emulsion surimi gels was also emphasized. The results may provide some constructive suggestions for the design and development of RTE surimi products with high thermal stability and desired gel properties.

## 2. Materials and Methods

### 2.1. Materials

Frozen cod surimi was purchased from Zhejiang Yufu Food Co., Ltd. (Hangzhou, China), and stored at −80 °C until use. Soybean oil was purchased from Cofco Fulinmen Food Marketing Co., Ltd. (Tianjin, China), and perilla seed oil was purchased from Hebei Jiafeng vegetable Oil Co., Ltd. (Hebei, China). The other chemical reagents were purchased from Shanghai Lingfeng Chemical Reagent Co., Ltd. (Shanghai, China).

### 2.2. Fatty Acid Composition

The weighed oil of 0.1 g was dissolved in 2 mL n-hexane, and transferred into a 5 mL centrifuge tube. Afterward, 0.5 mL of the NaOH-CH_3_OH solution was mixed for 2 min using a vortex shaker, and the upper organic phase was transferred to a 5 mL centrifuge tube. An appropriate amount (about 2 g) of anhydrous sodium sulfate was then added for dehydration before shaking and centrifuging (8000× *g*, 5 min) [15].

Gas chromatography equipped with a flame ionization detector (7890A, Agilent Technologies, San Francisco, CA, USA) was employed to determine the fatty acid composition of the two vegetable oil samples. A DB-17MS capillary column (30 m × 0.25 mm × 0.25 μm) was used with He as the carrier gas (purity 99.999%) at the flow rate of 1 mL/min. The sample passed through a 0.22 μm organic membrane, and a 1 μL injection was administrated. The split ratio was 10:1, and the sample collection time was 30 min. The inlet and detector temperatures were both set at 260 °C. The temperature of the FID detector increased from initial 130 °C to 190 °C at 5 °C/min for 2 min and then increased to 240 °C at 5 °C/min for 10 min [16].

### 2.3. Extraction of Myofibrillar Proteins

The myofibrillar protein extraction was performed as described by Kim et al. [17]. In brief, two parts of 0.58 mol/L saline (0.49 mol/L NaCl, 17.8 mmol/L Na_5_P_3_O_10_, and 1 mmol/L NaN_3_, pH 8.3, 2 °C) solution was mixed with one part of ground meat for 1 h. The slurry was then centrifuged at 12,000× *g* at 4 °C for 1 h. The myofibrillar protein extract was finally strained through layers of cheesecloth.

### 2.4. Preparation of Pre-Emulsified Oils

A homogeneous test was performed with a ULTRA-TURRAX (IKA. T25 digital, Berlin, Germany). The various pre-emulsification ratios of 20%, 40%, 60% and 80% of perilla seed oil and soybean oil were obtained by emulsifying the different amounts (0.6 g, 1.2 g, 1.8 g and 2.4 g) of oils, named as P1, P2, P3, and P4, and S1, S2, S3, and S4, respectively. The P0 and S0 were the control group of perilla seed oil and soybean oil without any emulsification treatments. In detail, two types of oils were emulsified by 10 mL-myofibrillar protein solution of 1 mg/mL under homogenization for 1 min at 10,000 r/min. The overall addition amount (3 g) of oil was introduced into surimi (100 g) for subsequent preparation of surimi gel (Section 2.8).

### 2.5. Droplet Diameter

The droplet diameter of emulsions was measured at room temperature using a OLYMPUS BX41TF instrument. The emulsion prepared above was dispersed in 0.05% coomassie blue containing 0.1% SDS immediately, and then was observed by an optical microscope equipped with a digital camera [18]. Magnification was 100 times.

### 2.6. Particle Size and Distribution

The particle size of emulsions was measured at room temperature using a Zetasizer Nano-ZS900 instrument (Malvern Instruments Co. Ltd., London, UK). Measurements were performed in three replicates [19].

### 2.7. Emulsifying Activity (EAI) and the Emulsifying Stability Index (ESI)

After homogenization, 20 µL of each emulsion was diluted with 5 mL of 0.1% (*w*/*v*) SDS solution and shaken rapidly. The absorbance was measured at 500 nm [20].

The following Equations (1) and (2) were used for calculation of EAI and ESI, respectively.
EAI (m^2^·g^−1^) = 2 × 2.303 × A_0_ × N/(C × Φ × 10^4^)(1)
ESI (min) = A_0_/(A_0_ − A_10_) × T(2)

N = dilution factor (1000); C = protein concentration (mg/mL); Φ = oil volume fraction; T = 10 min; A_0_ = absorbance at 0 min; and A_10_ = absorbance at 10 min.

### 2.8. Preparation of Emulsion Surimi Gel

Ten pieces of frozen surimi of 100 g were cut into about 1 cm × 1 cm × 1 cm after half thawing. Then, 2% (*w*/*w*) salt, emulsion and fat were put into the beating machine. The samples without oil emulsion were regarded as control group. The moisture content of the final mixture in 10 groups was controlled at 78%, and the total amount of fat (perilla seed oil and soybean oil) accounted for 3% of surimi quality. The mixed sample was chopped at 3000 r/min for 5 min (below 8 °C). The chopped fish pulp was poured into the collagen casing (22 mm in diameter). After the bubbles were drained, the two ends were tied tightly. A two-stage heating method was used (40 °C for 1 h and 90 °C for 20 min), and the resultant surimi gels were placed in an autoclave for sterilization at 115 °C for 10 min, then cooled immediately and stored in a refrigerator at 4 °C until measurements [14].

### 2.9. Color

Whiteness was measured using a colorimeter (Color Quest XE, Tokyo, Japan) [21]. The *L** (lightness), *a** (redness), and *b** (yellowness) were obtained by eight replicate measurements for each sample. The whiteness (*W*) was calculated using the following Equation (3):*W* = 100 − [(100 − *L**)^2^ + *a**^2^ + *b**^2^]^1/2^(3)

### 2.10. Gel Strength

A penetration test was performed with a texture analyzer (TA. XT Plus, Los Angeles, CA, USA) [12]. Surimi samples were equilibrated at room temperature (about 25 °C) for 2 h prior to the gel strength. Samples were cut into cylinders (22 mm in diameter and 20 mm in height). The test conditions were as follows, probe: P/5S; pre-test speed: 1.00 mm/s; test speed: 2.00 mm/s; post-test speed: 10.00 mm/s; displacement: 15 mm; triggering mode: automatic (power); trigger force: 10.0 g. Measurements were performed in eight replicates.

### 2.11. Water Holding Capacity

The determination of water holding capacity for the samples was measured by centrifuge [12]. Gel samples were cut into thin slices, and approximately 2 g samples were weighed and placed between two layers of filter paper. Subsequently, the samples were placed at the bottom of centrifuge tubes and centrifuged at 10,000× *g* for 15 min. After centrifugation, gels were weighed again. Measurements were performed in eight replicates.
WHC (%) = W_2_/W_1_ × 100(4)
where W_1_ refers to initial weight of gels (g), W_2_ refers to final weight of gels (g).

### 2.12. Texture Analysis

Texture analysis was performed with a texture analyzer (TA. XT Plus, USA) [21]. Samples were cut into cylinders (22 mm in diameter and 20 mm in height) and equilibrated at room temperature (about 25 °C) for 2 h. TPA test conditions were as follows, probe: P/36R; pre-test speed: 1.00 mm/s; test speed: 2.00 mm/s; post-test speed: 10.00 mm/s; strain: 50%; triggering mode: automatic (power); trigger force 10.0 g. Measurements were performed in eight replicates.

### 2.13. Rheological Properties

The rheological properties were determined by a rheometer (MCR302, Anton Paar Ltd., Sydney, Austria). A slit of 1 mm was set and silicone oil was used to prevent water evaporation. The storage modulus (G′) was measured with the dynamic temperature sweep from 20 °C to 120 °C was performed at the heating rate of 4 °C/min at 0.1 Hz [22].

### 2.14. Determination of Protein Solubility

The weighted chopped gel (2 g) was homogenized for 1 min with 10 mL 0.05 mol/L NaCl solution (SA), 0.6 mol/L NaCl solution (SB), 0.6 mol/L NaCl + 1.5 mol/L urea solution (SC), 0.6 mol/L NaCl + 8 mol/L urea solution (SD), and 10 mL 0.6 mol/L NaCl + 8 mol/L urea + 0.05 mol/L *β*-mercaptoethanol solution (SE), respectively. The samples were placed at 4 °C for 1 h and then centrifuged for 15 min at 10,000× *g*. Subsequently, the coomassie brilliant blue method was used to determine the protein content of the supernatant [23]. The index of ionic bands, index of hydrogen bonds, index of hydrophobic interactions and index of disulfide bond were expressed as follows: index of ionic bands = SB − SA;index of hydrogen bonds = SC − SB;index of hydrophobic interactions = SD − SC;index of disulfide bond = SE − SD.

### 2.15. Statistics Analysis

The data were analyzed by SPSS 17.0 software. Duncan multiple test was used to determine the significance between the data at *p* < 0.05 level. All values were presented as means ± standard deviation.

## 3. Results and Discussion

### 3.1. Fatty Acid Composition

Five fatty acids including saturated fatty acids (palmitic acid and stearic acid), as well as unsaturated fatty acids (oleic acid, linoleic acid and linolenic acid) with the different percentages are found in both perilla seed oil and soybean oil (Table 1). Perilla seed oil has the highest linolenic acid percentage of 54.44% in fatty acid composition, while linoleic acid accounts for the highest proportion (43.82%) in soybean oil. Besides, it also can be observed that perilla seed oil has the higher proportion of unsaturated fatty acids (90.09%) than that of soybean oil (82.03%). The respective fatty acid compositions of two types of oils are consistence with the previous studies [24].

### 3.2. Droplet Size and Distribution

The droplet diameter and particle size distribution reflect homogeneity of fat spheres in the emulsion system, which further reveal the stability of the emulsion system. The droplet diameter and size distribution of perilla seed oil and soybean oil with the different emulsifying ratios are shown in Figure 1. An increasing droplet size of emulsion is found for two types of oils along with the rising pre-emulsification ratio (Figure 1A). For example, the average particle size of perilla seed oil and soybean oil emulsion increases from 821.62 nm and 464.29 nm (20% emulsification ratio) to 2746.31 and 1155.43 nm (40% emulsification ratio), respectively (Figure 1B). However, two kinds of emulsions have a similar size distribution with the peaks at 220.25 nm and 1005.73 nm (Figure 1C). The increasing emulsification ratio means more available surfactants are required to further reduce the droplet size of the emulsions. However, the insufficient amounts of stabilizers may difficultly meet the requirements of increasing interfacial area, thus leading to the formation of larger droplets. It is worth noting that the droplet size of perilla seed oil emulsions greatly depends on emulsifying ratios relatively to soybean oil emulsions. In particular, a sharp increase in mean droplet size is observed for perilla seed oil with emulsifying ratio of 60% (P3 group). Additionally, the mean particle size of perilla seed oil emulsion is significantly higher than that of soybean oil with the same emulsification ratio. The difference in droplet size of two emulsions may be ascribed to their distinct fatty acid composition. The perilla seed oil has a higher unsaturation degree of fatty acid than soybean oil (Table 1). Previous studies have also consolidated that the droplet size distribution of emulsified fatty acids is largely determined by saturation degree of fatty acids [10]. The drop coalescence and flocculation between the fat particles can be prevented by the resulting interfacial protein membranes from adsorbed salt-soluble proteins. Han et al. [25] found that the low unsaturated fatty acids were conducive to the adsorption and diffusion of salt-soluble proteins, effectively preventing the drop coalescence and flocculation of fat particles. Inversely, the fatty acids with high unsaturation degree would weaken the binding ability with interfacial protein membrane due to the reduced hydrophobic interactions, resulting in coalescence of fat particles [26].

### 3.3. Emulsifying Properties

Emulsifying activity depends on protein–lipid and protein–protein interactions, while emulsifying stability is affected by the various factors associating with the continuous and dispersed phases [27]. The quality of emulsified meat products can be assessed by these two important indicators. The effects of oil types and emulsifying ratio on EAI and ESI are shown in Figure 2. The rising pre-emulsification ratio leads to the increase of EAI and decrease of ESI for two kinds of emulsions. The results may be ascribed to the amounts of proteins surrounding the oil/water interface, which is one of stabilization mechanisms in emulsified meat products. The higher pre-emulsification ratio is achieved by the increasing oil globules in the given protein emulsifier systems, which decreases ESI by reducing interfacial film thickness. Besides, the increase of emulsion particle size (Figure 1) also causes the poor emulsion stability. EAI refers to the emulsifying activity of emulsifier per unit weight. In this work, the different amounts of oils were emulsified by the given amounts of myofibrillar protein, and thereafter the various pre-emulsification ratios were obtained. The results suggest that rising pre-emulsification ratio leads to the increase of EAI for two kinds of emulsions. The reason may be no saturated emulsifying status is achieved for the present dosage of protein. We speculate that a subsequent decrease of EAI may be observed for the emulsion with the higher pre-emulsification ratios (>0.8), due to less availability of protein. Some differences in EAI and ESI are observed for both two emulsions. The large droplets sizes of perilla seed oil relatively to soybean oil may be responsible for their stability difference [28]. The difference in the EAI of emulsion are closely related to their distinct protein adsorption ability [25]. The higher unsaturation degree of perilla seed oil confers its enhanced binding ability with proteins, which further improves the emulsifying activity of emulsions.

### 3.4. Gel Properties

Effects of oil types and emulsifying ratio on the gel properties of surimi subjected to heating treatments are shown in Figure 3A. Soybean oil emulsion gel shows the relatively greater whitening effect, compared to perilla seed oil emulsion. The less pigmentation and higher brightness of soybean oil may be responsible for the color differences [14]. The increasing pre-emulsification ratio causes the whiteness improvement of surimi gel, and the highest whiteness is obtained when the pre-emulsification ratio reaches 80%. Such whitening effect of vegetable oil is ascribed to the light scattering effect of oil droplets in surimi gel [29].

The gel strength and water retention are important quality indicators, which determine the popularity of surimi products. The thermal stability of emulsion gels is reflected by gel properties (gel strength, water retention and texture properties) of surimi subjected to heating treatments. Effects of the pre-emulsification ratios and oil types on gel strength and water retention of two emulsion gels subjected to heating treatments are shown in Figure 3B. The gel strength and water retention of both two kinds of surimi gel display a firstly increased and then decreased tendence along with the rising pre-emulsification ratios. Moreover, the emulsion gels display the higher thermal stability than the control group regardless of oil types. The similar results are also reported in the recent studies [8]. The initial increase in gel strength is ascribed to introduction of emulsifying system. On one hand, the rising addition of myofibrillar pre-emulsified oil contributes to the enhanced gel strength of surimi to some extent by the incorporation of proteins [30]. On the other hand, the oil surface is coated with protein membrane to form fine oil droplets after pre-emulsification, which further induces the formation of interfacial protein films via interaction with the hydrophobic regions of myofibrillar protein [31]. The resultant interfacial protein films can effectively prevent the polymerization and flocculation between fat particles. Gel strength and water retention largely depend on the gel microstructure (Appendix A) [32]. The oil globules entrapped in the matrix induce the formation of the denser and homogeneous gel structure, which endows the great mechanical strength and water-holding capacity. However, the size of oil globules increases along with the higher pre-emulsification ratio (40%), which retards the formation of the ordered gel network structure to some extent by weakening the protein-water interaction [14]. Specially, the pre-emulsification ratios where gel strength and water retention obtain the greatest value are different for the two types of surimi. For example, the addition of perilla seed oil with pre-emulsification ratio of 20% (P1 group) allows the highest gel strength, while pre-emulsification ratio of soybean oil delays to 40% (S2 group). This is closely related to the droplet diameter. The similar droplet diameter of P1 as S2 can be evenly dispersed in the internal structure of the gel, increasing the compactness and uniformity of the three-dimensional network structure of surimi gel (Appendix A). Moreover, the emulsion surimi with perilla seed oil always shows the higher gel strength and water retention than that with soybean oil under the same pre-emulsification ratio. The difference in gel strength and water retention may be ascribed to the unsaturation degree. The high surface hydrophobicity (high unsaturation degree) may facilitate the favorable crosslink between protein–protein and protein–fat to form the better interfacial protein films, which benefits for the stable gel properties [31].

The effects of perilla seed oil and soybean oil with different pre-emulsification ratios on texture properties (hardness and chewability) of surimi gel are displayed in Figure 3C,D. An initial increase in hardness and chewiness of emulsion surimi gels is observed with rising of pre-emulsification ratio until 20% (P1) or 40% (S2), where texture properties reach the highest value. For example, the gel hardness and chewability of perilla seed oil and soybean oil emulsion gels increase by 33.10%, 57.91% and 28.17%, 39.54% compared to control group, respectively. However, hardness and chewiness of surimi gels remain stable when the pre-emulsification ratio exceeds over 40%. Such a nonlinear tendency of texture profiles as a function of the pre-emulsification ratio may be ascribed to the filling effects of oil globules, which are supported by the previous reports [14]. Oil globules with pre-emulsification treatment are filled in the protein matrix by physical restrictions, contributing to the formation of gel network with a denser structure (Appendix A). Besides, the resultant interfacial protein films from dispersed oil globules and the hydrophobic regions of myofibrillar protein also benefit the development of a denser microstructure. The homogeneous gel structure endows the great mechanical strength to resist deformation [9]. When pre-emulsification ratio exceeds over 40%, the larger oil droplets retard the formation of the ordered gel network structure to some extent by weakening the protein–water interaction (Appendix A) [33].

### 3.5. Rheological Properties

The dynamic temperature sweep curves ranging of 20~120 °C of surimi gel emulsified by perilla seed oil and soybean oil are shown in Figure 4. In general, a similar change of storage modulus (G′) as a function of temperature is found for two types of emulsion gels. The initial enhancement of the G′ indicates the formation of a weak gel network, and the greatest G′ value is achieved when heated to 45 °C. The heating treatments induce the exposure of active groups, which further promote the aggregation of myosin head [34]. A sharp decrease in myosin tail expansion temporarily deteriorates the gel network structure. The sharp increase of G′ after 52 °C suggests the formation of thermally irreversible gel structures [35]. However, the G′ magnitude displays a significant difference in surimi gels with the different pre-emulsification ratios. The G′ values of both two emulsion gels display a firstly increased and then decreased tendency along with the increasing pre-emulsification ratios. The G′ reaches the peak value when the pre-emulsification ratio is 20% and 40% for perilla seed oil and soybean oil, respectively. However, the further rising pre-emulsification ratio does not cause a continuously increase of the G′ value. The results are also consistent with gel strength and texture results. The pre-emulsified oil globules allow for the formation of the denser and homogeneous gel structure modulated by interfacial protein films, which is responsible for the enhanced dynamic modulus (Appendix A). However, the formation of the ordered gel network structure is retarded due to lager oil droplets by weakening the protein–water interaction. The G′ has two maximum values, which are closely related to the types of oil and the composition of fatty acids. Due to the higher proportion of unsaturated fatty acids, the structure transitions were conspicuously observed in the surimi systems using perilla seed oil during heating.

### 3.6. Protein Solubility

The ionic bonds, hydrogen bonds, hydrophobic interactions and disulfide bonds support the three-dimensional network integrity of surimi gel in the different ways. It is thereafter of great significance for underlying thermal stability mechanisms by the insights into the various interactions. determine the technological properties of surimi products [36]. Ionic bonds mainly stabilize the tertiary and quaternary structure of proteins, while hydrogen bonds are responsible for secondary structure of proteins. The resultant disulfide bonds from oxidation of sulfhydryl groups support the formation of protein spatial structures. The aggregation protein molecular is mainly mediated by hydrophobic interactions [37,38]. The effects of oil types and emulsification ratio on the protein solubility of surimi gel are shown in Figure 5. The increase of pre-emulsification ratio causes a firstly increased and then decreased change for hydrogen bond, ionic bond and hydrophobic interaction. The peak value is achieved when the pre-emulsification ratio of perilla seed oil and soybean oil is 20% and 40%, respectively. The similar change to gel properties also suggest that these interaction forces are the inherent contributors. The oppositive results are observed for changes of disulfide bond. The reasons may be the introduction of emulsion systems induces the changes of protein conformation and the corresponding interaction forces [25]. Meanwhile, the disulfide bonds are formed by the oxidation of sulfhydryl groups, which reflect the structural stability of myofibrillar protein. The results reveal that the sulfhydryl groups of myofibrillar protein are not liable to oxidation under the pre-emulsification ratio. The higher structure stability of myofibrillar protein is ascribed to the greater other interactions (hydrogen bonds and hydrophobic interactions), which are also well regarded as the main contributors associating with the structure stability of protein. The introductions of interfacial protein membrane embedding oil droplets contribute to the enhanced hydrogen bonds. The rising pre-emulsification treatments also cause more exposure of hydrophobic groups, and thereby improve hydrophobic interaction. The higher pre-emulsification ratio is achieved by the increasing oil globules in the given protein emulsifier systems. The insufficient amounts of proteins may meet the requirements of increasing interfacial area with difficulty, thus leading to the formation of larger oil droplets. The hydrophobic binding sites of oil may be difficultly exposed due to the drop coalescence, which decreased the hydrophobic interaction with protein. These resultant lager oil droplets from over-emulsification treatments may fail to stabilize emulsion gel systems due to the weakening hydrophobic interactions, which further affect the structure integrity evidenced by increased disulfide bonds [6].

## 4. Conclusions

The introduction of emulsion has been proven as a valid method for the preparation of surimi products with the improved quality. In the present work, emulsion-formulated surimi gels were prepared to enhance the resistance to heating treatments. The results show that the thermal stability of emulsion gels was improved, evidenced by the changes of gel properties of surimi after heating treatments. The gel properties of both kinds of surimi gels can be modulated via the oils and the rising pre-emulsification ratios. The gel strength, water retention, elastic modulus and texture properties of both kinds of surimi gels display a firstly increased and then decreased tendency with the rising pre-emulsification ratios. In general, perilla seed oil emulsion surimi gels with an emulsification ratio of 20% (P1 group) contribute to the improved thermal stability, compared to the soybean oil emulsion gels.

## Figures and Tables

**Figure 1 foods-11-00179-f001:**
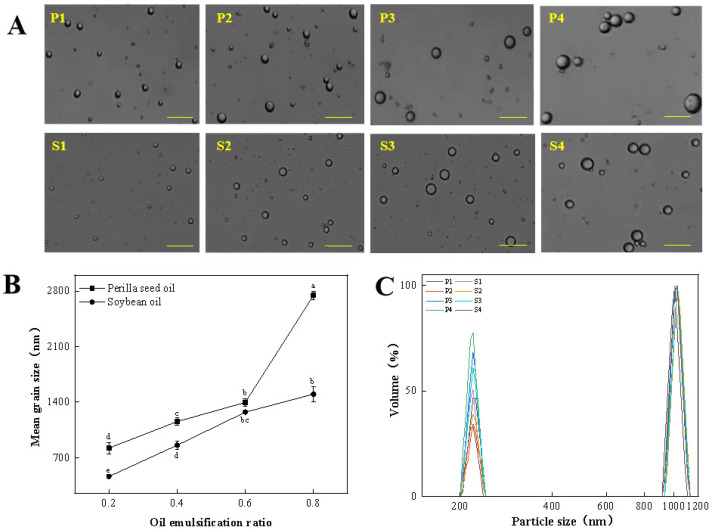
The droplet diameter (**A**), mean grain size (**B**) and particle size distribution (**C**) of perilla seed oil and soybean oil with the different emulsifying ratios (The measurement was done at backscatter angle of 173° and at room temperature, and the particle size and distribution were obtained using the built-in analytical software of Zetasizer Nano-ZS900 instrument). Scale bar: 100 μm. (Different little letters mean a significant difference at the level of 0.05, *p* < 0.05).

**Figure 2 foods-11-00179-f002:**
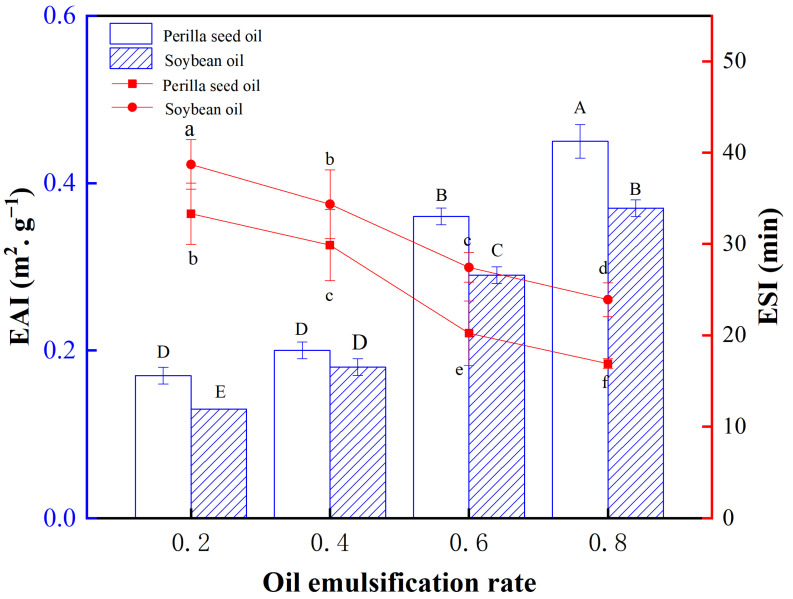
The EAI and ESI of perilla seed oil and soybean oil with the different emulsifying ratios. (Different capital letters or little letters mean a significant difference at the level of 0.05, *p* < 0.05).

**Figure 3 foods-11-00179-f003:**
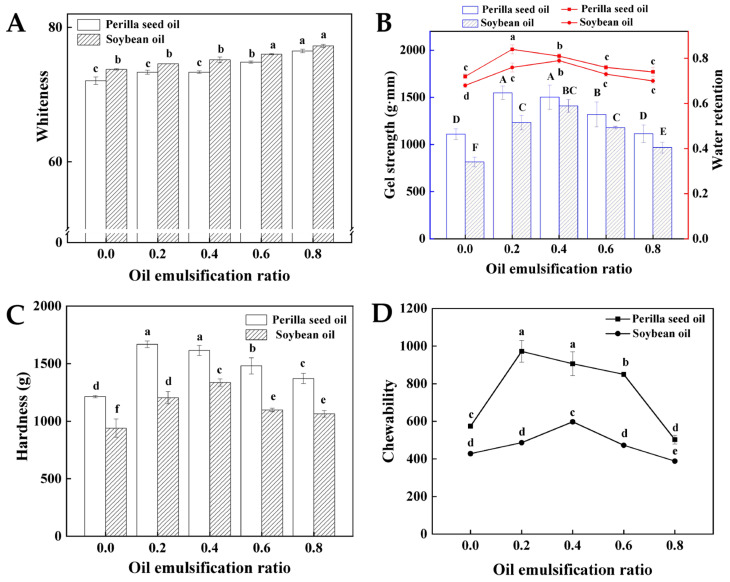
The gel properties of surimi gel emulsified by perilla seed oil and soybean oil subjected to heating treatments (**A**) whiteness, (**B**) gel strength and water retention, (**C**) hardness and (**D**) chewability. (Different capital letters or little letters mean a significant difference at the level of 0.05, *p* < 0.05).

**Figure 4 foods-11-00179-f004:**
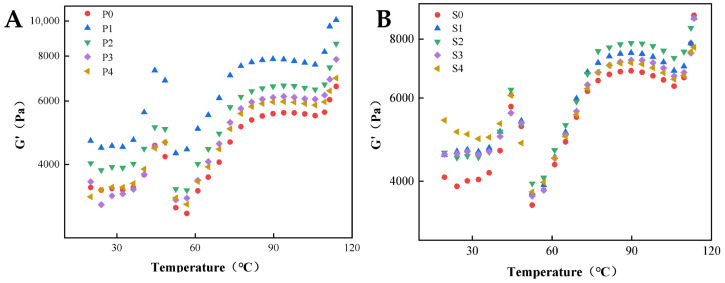
The dynamic temperature sweeps of surimi gel emulsified by perilla seed oil (**A**) and soybean oil (**B**).

**Figure 5 foods-11-00179-f005:**
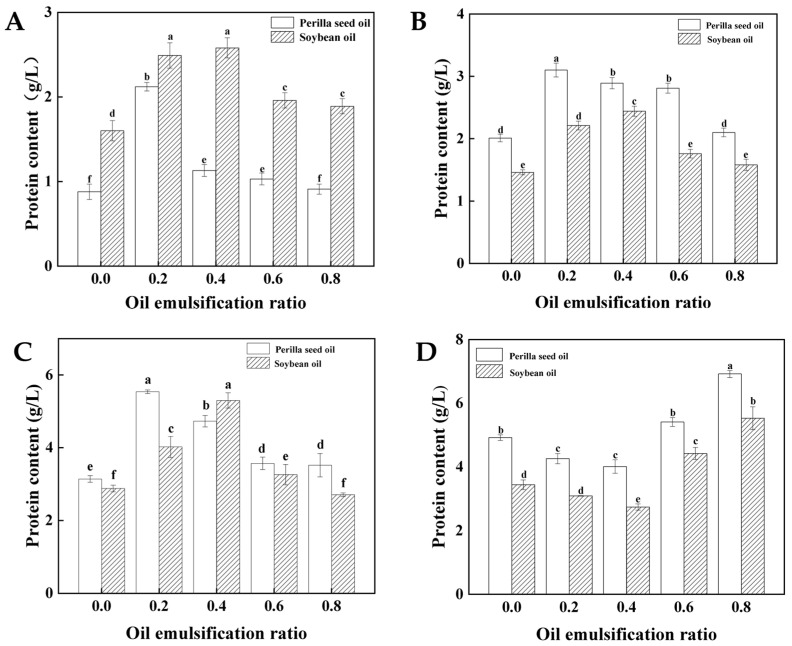
The protein solubility of surimi gel emulsified by perilla seed oil and soybean oil subjected to heating treatments (**A**) ionic bonds, (**B**) hydrogen bonds (**C**) hydrophobic interactions and (**D**) disulfide bonds. (Different little letters mean a significant difference at the level of 0.05).

**Table 1 foods-11-00179-t001:** Fatty acid composition of perilla seed oil and soybean oil.

	Palmitic Acid	Stearic Acid	Oleic Acid	Linoleic Acid	Linolenic Acid
Perilla seed oil	8.51%	1.11%	19.83%	15.82%	54.44%
Soybean Oil	13.66%	2.03%	31.77%	43.82%	6.44%

## Data Availability

Not applicable.

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
