# Peer review of "Emulsion Surimi Gel with Tunable Gel Properties and Improved Thermal Stability by Modulating Oil Types and Emulsification Degree"

_foods, 2022, doi:10.3390/foods11020179_

Round 1

Reviewer 1 Report

2.4. Preparation of pre-emulsified oils: The homogenizer model should be specified.
2.5. Droplet diameter: Please, specify the model of microscope and magnification.
2.13 Rheological properties: What kind of measuring system was used? Cone-to-plate? Plate-to-plate?
3.3 Emulsifying properties: It will be useful to measure interface tension on the water/oil interface for perilla seed oil and soybean oil to a better understanding of their emulsifying properties.
3.5. Rheological properties: Figure 4. It is better to use a log scale on the y-axis of a plot.

Author Response

Response to reviewer’ comments

Dear  reviewer:

Thank you so much for the time and meaningful comments. The manuscript has been carefully revised according to reviewers’ comments and also the language of the whole manuscript has been carefully revised. Below are our responses to the reviewers’ comments. Besides, amendments in the revised manuscript have been highlighted.

Question 1. 2.4. Preparation of pre-emulsified oils: The homogenizer model should be specified.

Response: Many thanks for your constructive comments. A homogeneous test was performed with a ULTRA-TURRAX (IKA. T25 digital, Germany).

  1. 5. Droplet diameter: Please, specify the model of microscope and magnification.

Response: The droplet diameter of emulsions was measured at room temperature using OLYMPUS BX41TF instrument. Magnification was 100 times. Many thanks for your constructive comments.

  1. 13.Rheological properties: What kind of measuring system was used? Cone-to-plate? Plate-to-plate? 

Response: The measuring system of plate-to-plate was used. Thanks

  1. Emulsifying properties: It will be useful to measure interface tensionon the water/oil interface for perilla seed oil and soybean oil to a better understanding of their emulsifying properties.

Response: Many thanks for your constructive comments. The interface tension is a valid indicator to reflect the emulsifying properties of emulsion. However, interface tension measurements are difficultly performed for us due to the lack of the corresponding equipment. In this work, emulsifying activity index (EAI) and emulsifying stability index (ESI) were used to characterized the emulsifying properties, which are also widely used methods by the previous studies. For example, Aziz et al. (Effect of protein and oil volume concentrations on emulsifying properties of acorn protein isolate. Food Chem. 2020) analyzed the emulsification behavior of rubber protein isolate at oil-water interface by studying EAI, ESI and droplet size. Guo et al. (Emulsifying properties of sweet potato protein: Effect of protein concentration and oil volume fraction. Food Hydrocoll. 2011) studied that the effect of protein concentrations and oil volume fractions on properties of stabilized emulsions of sweet potato proteins (SPPs) were investigated by use of the EAI, ESI and droplet size et al.

  1. Rheological properties: Figure 4. It is better to use a log scale on the y-axis of a plot.

Response: Thanks for your comments. The revised Figure 4 was seen as follows:

Reviewer 2 Report

Review of “Emulsion surimi gel with tunable….” By Zhu et. al.

The authors have investigated the effect of oil type and emulsification ratio on the physical properties of surimi gel. The emulsification properties of the two oils before gelation were characterized via EAI and ESI. Mean drop size was found to be larger for Perilla seed oil and increased with emulsification ratio for both oils. It was difficult to discern the difference in drop size distribution from Fig. 1C. Interestingly, EAI increased with emulsification ratio whereas ESI decreased for both oils. Gel strength, hardness and chewability all exhibited a maximum at intermediate oil emulsification ratio. The temperature sweeps of G’ at different emulsification ratios displayed a complex behavior which is not fully explained in the manuscript. There are two maxima and sharp increases in G’ and G” at three temperatures. In addition, the maxima in G’ is exhibited for P1 for perilla oil whereas the maximum in G” occurs for S2 for soybean oil. This is not explained as well in the manuscript. Protein solubility also exhibits a maximum at intermediate oil emulsification ratio except of disulfide bonds. The methods are well described. The results are clearly presented. However, the discussion of the results is not satisfactory. As pointed out above and also in the following comments, several questions need better explanation. It is suggested that the authors provide these in their revision.
1.    Line 243, I presume by “polymerization”, the authors mean “drop coalescence”. If so, this should be replaced. This is used in several other instances in the manuscript. It is suggested that the authors make the corrections there as well.
2.    Line 248, the authors need to explain what they mean by “weaken the binding ability of interfacial protein membrane”.
3.    Line 247, “fat acid” should be “fatty acid”.
4.    Authors need to include a better particle size distribution plots in Fig. 1C.
5.    Lines 260 to 268 – The explanation given by authors for observed increased in EAI with emulsification ratio is not complete. One would expect EAI to decrease with more oil (higher ratio) since less protein will be available to cover the oil surface. This in turn will increase interfacial tension thereby making it harder to break the droplets leading to larger drop size and hence lower EAI (see the eq. for definition of EAI). Not clear as to why EAI increases with emulsification ratio. This needs to be rewritten.
6.    Lines 317 – 334 – the authors reiterate the observed behavior in Fig. 3 without offering an explanation. Please provide an explanation.
7.    Fig. 3B - why is the oil emulsification ratio at which maximum occurs for gel strength and water retention are different? Explain.
8.    Line 369 - not clear. Higher globule size has a lower surface area and hence higher surface concentration of proteins at the surface. needs a better explanation. How is the gel strength related to oil droplet size? As pointed out above, the authors need to provide a much better explanation for the observed rheological behavior and differences between the two oils.  Fig. 4 – why are there two maxima in G’- explain.
9.    Line 383 – the sentence “The aggregation protein mo-383 lecular is mainly mediated by hydrophobic interactions” …. And “ the thermal treatments 384 cause the microstructure deterioration to some extent by disrupting the weak molecular 385 forces (ionic bonds and hydrogen bonds). It is thereafter of great significance for under-386 lying thermal stability mechanisms by the insights into the various interactions.” are general statements. The authors need to make specific reference to the two systems investigated.
10.    Line 389 – the authors state “The increase of pre-emulsification ratio causes a firstly increased and 389 then decreased change for hydrogen bond, ionic bond and hydrophobic interaction.”. It is not clear why.
11.    Line 394- the sentence “The reasons may be the introduction of emulsion systems induces the changes of protein conformation and the corresponding interaction forces” is speculative. 
12.    Line 399 – the sentence “However, the resultant lager oil droplets from over emulsification treatments may fail to stabilize emulsion gel systems due to the weakening hydrophobic interactions”  is not clear. Needs a better explanation. 
13.    Fig. 5D – why is there a minimum in disulfide bonds unlike the other interactions? Explain. 
14.    It is suggested that the authors include some micrographs of surimi gel to demonstrate the difference in structure of gels for the two oils.

Author Response

Response to reviewer’ comments

Dear reviewer:

Thank you so much for the time and meaningful comments. The manuscript has been carefully revised according to reviewers’ comments and also the language of the whole manuscript has been carefully revised. Below are our responses to the reviewers’ comments. Besides, amendments in the revised manuscript have been highlighted.

Question 1. Line 243, I presume by “polymerization”, the authors mean “drop coalescence”. If so, this should be replaced. This is used in several other instances in the manuscript. It is suggested that the authors make the corrections there as well.

Response: Many thanks for your suggestions. Revised (L247, L251, L306).

  1. 2.Line 248, the authors need to explain what they mean by “weaken the binding ability of interfacial protein membrane”.

Response: Many thanks for your suggestions. The previous reports indicated that the fatty acids with high unsaturation degree retarded the diffusion and adsorption of salt-soluble proteins due to the reduced hydrophobic interactions (Han et al. Effects of fatty acid saturation degree on salt-soluble pork protein conformation and interfacial ad-sorption characteristics at the oil/water interface. Food Hydrocoll. 2021). We have revised the descriptions. “…the fatty acids with high unsaturation degree would weaken the binding ability with interfacial protein membrane due to the reduced hydrophobic interactions, resulting in coalescence of fat particles”

  1. Line 247, “fat acid” should be “fatty acid”.

Response: Many thanks for your suggestions. Revised (L247).

  1. Authors need to include a better particle size distribution plots in Fig. 1C.

Response: Many thanks for your suggestions. Revised in Fig. 1C.

  1. Lines 260 to 268 – The explanation given by authors for observed increased in EAI with emulsification ratio is not complete. One would expect EAI to decrease with more oil (higher ratio) since less protein will be available to cover the oil surface. This in turn will increase interfacial tension thereby making it harder to break the droplets leading to larger drop size and hence lower EAI (see the eq. for definition of EAI). Not clear as to why EAI increases with emulsification ratio. This needs to be rewritten.

Response: Many thanks for your suggestions. “EAI refers to the emulsifying activity of emulsifier per unit weight. In this work, the different amounts of oils were emulsified by the given amounts of myofibrillar protein, and thereafter the various pre-emulsification ratios were obtained. The results suggest that rising pre-emulsification ratio leads to the increase of EAI for two kinds of emulsions. The reason may be no saturated emulsifying status is achieved for the present dosage of protein. We speculate that a subsequent decrease of EAI may be observed for the emulsion with the higher pre-emulsification ratios (>0.8), due to less availability of protein”. Thanks again.

  1. Lines 317 – 334 – the authors reiterate the observed behavior in Fig. 3 without offering an explanation. Please provide an explanation.

Response: Many thanks for your suggestions. The explanations have been added in Line 317 – 334. “Specially, the pre-emulsification ratios where gel strength and water retention obtain the greatest value are different for two types of surimi. For example, the addition of perilla seed oil with pre-emulsification ratio of 20 % (P1 group) allows the highest gel strength, while pre-emulsification ratio of soybean oil delays to 40 % (S2 group). This is closely related to the droplet diameter. The similar droplet diameter of P1 as S2 can be evenly dispersed in the internal structure of the gel, increasing the compact-ness and uniformity of the three-dimensional network structure of surimi gel. Moreover, the emulsion surimi with perilla seed oil always shows the higher gel strength and water retention than that with soybean oil under the same pre-emulsification ratio. The difference in gel strength and water retention may be ascribed to the unsaturation degree. The high surface hydrophobicity (high unsaturation degree) may facilitate the favorable crosslink between protein-protein and protein-fat to form the better interfacial protein films, which benefits for the stable gel properties [31].”    

  1. 3B - why is the oil emulsification ratio at which maximum occurs for gel strength and water retention are different? Explain.

Response: Many thanks for your suggestions. We have confirmed that the oil emulsification ratio at which maximum occurs for gel strength and water retention are consistent for the same oil emulsion. The different pre-emulsification ratios where gel strength and water retention obtain the greatest value for two types of emulsified gels may be ascribed to their distinct droplet diameter (Figure 1). The mean particle size of perilla seed oil emulsion is always higher than that of soybean oil due to their different fatty acid composition. The smaller droplets of perilla seed oil than soybean oil can be evenly dispersed in the internal structure of the gel, increasing the compactness and uniformity of the three-dimensional network structure of surimi gel. The physical filling effects contribute to an overall increase in the water holding capacity of surimi gel. While the oil droplets with larger diameter retards the formation of the ordered gel network structure to some extent by weakening the protein-water interaction.

  1. Line 369 - not clear. Higher globule size has a lower surface area and hence higher surface concentration of proteins at the surface. needs a better explanation. How is the gel strength related to oil droplet size? As pointed out above, the authors need to provide a much better explanation for the observed rheological behavior and differences between the two oils. Fig. 4 – why are there two maxima in G′- explain.

Response: Many thanks for your suggestions. The introduction of oil droplets with the changing sizes into surimi can affect the gel strength by modulating the microstructure of gel (Figure 1S). The oil globules with the small size allow for the formation of the denser and homogeneous gel structure modulated by interfacial protein films, which is responsible for the enhanced dynamic modulus. However, the formation of the ordered gel network structure is retarded due to lager oil droplets by weakening the protein-water interaction. The similar results are also found in some other studies (Physicochemical properties and microstructure of fish myofibrillar protein-lipid composite gels: Effects of fat type and concentration. Food Hydrocoll. 2019). The gelation behavior of surimi can explain the G′ changes in Fig. 4. The initial enhancement of the G′ indicates the formation of a weak gel network, and the greatest G′ value is achieved when heated to 45 °C. The heating treatments induce the exposure of active groups, which further promote the aggregation of myosin head (Conformational changes and dynamic rheological properties of fish sarcoplasmic proteins treated at various pH. Food Chem. 2010). A sharp decrease in myosin tail expansion temporarily deteriorates the gel network structure. The sharp increase of G′ after 52 °C suggests the formation of thermally irreversible gel structures (Textural and viscoelastic properties of pork frankfurters containing canola-olive oils, rice bran, and walnut).

  1. Line 383 – the sentence “The aggregation protein molecular is mainly mediated by hydrophobic interactions” …. And “ the thermal treatments 384 cause the microstructure deterioration to some extent by disrupting the weak molecular 385 forces (ionic bonds and hydrogen bonds). It is thereafter of great significance for under-386 lying thermal stability mechanisms by the insights into the various interactions.” are general statements. The authors need to make specific reference to the two systems investigated.

Response: Many thanks for your suggestions. We have made the more detailed explanations on the interaction changes of surimi gel along with the emulsion treatments. “The increase of pre-emulsification ratio causes a firstly increased and then decreased change for hydrogen bond, ionic bond and hydrophobic interaction. The peak value is achieved when the pre-emulsification ratio of perilla seed oil and soybean oil is 20 % and 40 %, respectively. The similar change to gel properties also suggest that these interaction forces are the inherent contributors. The oppositive results are observed for changes of disulfide bond. The reasons may be the introduction of emulsion systems induces the changes of protein conformation and the corresponding interaction forces [25]. The introductions of interfacial protein membrane embedding oil droplets contribute to the enhanced hydrogen bonds. The rising pre-emulsification treatments also cause the more exposure of hydrophobic groups, and thereby improve hydrophobic interaction. However, the hydrophobic binding sites of oil may be difficultly exposed due to the drop coalescence, which decreased the hydrophobic interaction with protein. These resultant lager oil droplets from over emulsification treatments may fail to stabilize emulsion gel systems due to the weakening hydrophobic interactions, which further affect the structure integrity evidenced by increased disulfide bonds [6].”

  1. Line 389 – the authors state “The increase of pre-emulsification ratio causes a firstly increased and 389 then decreased change for hydrogen bond, ionic bond and hydrophobic interaction.”. It is not clear why.

Response: Many thanks for your suggestions. We have made the more detailed explanations. “ The introductions of interfacial protein membrane embedding oil droplets contribute to the enhanced hydrogen bonds. The given pre-emulsification treatments also cause the more exposure of hydrophobic groups, and thereby improve hydrophobic interaction. However, the continuous introduction of emulsion systems induces the changes of protein conformation and the corresponding interaction forces [25]. The hydrophobic binding sites of oil may be difficultly exposed due to the drop coalescence along with the higher pre-emulsification ratio, which decreased the hydrophobic interaction with protein”.

  1. Line 394- the sentence “The reasons may be the introduction of emulsion systems induces the changes of protein conformation and the corresponding interaction forces” is speculative. 

Response: Many thanks for your suggestions. Previous studies have found that the conformation transformation of protein can be induced by unsaturated fatty acids. For example, Han et al concluded that the unsaturated fatty acids caused the exposure of more tryptophan residues, thus forming the polar environment, which facilitated the changes in protein conformation and their interaction (Han et al. Effects of fatty acid saturation degree on salt-soluble pork protein conformation and interfacial adsorption characteristics at the oil/water interface. Food Hydrocoll. 2021, 113.). In the present work, the changes in the various interactions between protein- protein and protein-fat were emphasized. Thanks again.

  1. Line 399 – the sentence “However, the resultant lager oil droplets from over emulsification treatments may fail to stabilize emulsion gel systems due to the weakening hydrophobic interactions”  is not clear. Needs a better explanation. 

Response: Many thanks for your suggestions. We have made more detailed explanations. “ The higher pre-emulsification ratio is achieved by the increasing oil globules in the given protein emulsifier systems. The insufficient amounts of proteins may difficultly meet the requirements of increasing interfacial area, thus leading to the formation of larger oil droplets. The hydrophobic binding sites of oil may be difficultly exposed due to the drop coalescence, which decreased the hydrophobic interaction with protein” .

  1. 5D – why is there a minimum in disulfide bonds unlike the other interactions? Explain. 

Response: Many thanks for your suggestions. “ The disulfide bonds are formed by the oxidation of sulfhydryl groups, which reflect the structural stability of myofibrillar protein. Other interactions including hydrogen bonds and hydrophobic interactions exactly reach a maximum at which disulfide bonds are minimum. The results reveal that the sulfhydryl groups of myofibrillar protein are not liable to oxidation under the pre-emulsification ratio. The higher structure stability of myofibrillar protein is ascribed to the greater other interactions (hydrogen bonds and hydrophobic interactions), which are also well regarded as the main contributors associating with the structure stability of protein”.

  1. It is suggested that the authors include some micrographs of surimi gel to demonstrate the difference in structure of gels for the two oils.

Response: Many thanks for your suggestions. We have performed the confocal laser scanning microscope of emulsified gel to demonstrate the difference in structure (Figure 1S).

Reviewer 3 Report

Dear Authors, 

Please, provide these informations:

  • Lines 19-23 and 24: please, explain EAI, ESI P1 and S1.
  • Paragraph 2.3. You extract the myofibrillar proteins. Could you please provide an SDS-gel analysis?
  • Paragraph 2.7. Please, list the equations.
  • Table 1. What do you mean by 3 and 2 on the left, close to the name of the oil?
  • Figure 1 C. Please, use the complete DLS figure and don't cut the "base". 
  • Figure 4 and 5. Please, adjust the scale. The scale is not the same in the graphs. By consequence, re-write the paragraph 3.5 and 3.6 based on this. 

Thanks for your corrections. 

Best, 

Author Response

Response to reviewer’ comments

Dear reviewer:

Thank you so much for the time and meaningful comments. The manuscript has been carefully revised according to reviewers’ comments and also the language of the whole manuscript has been carefully revised. Below are our responses to the reviewers’ comments. Besides, amendments in the revised manuscript have been highlighted.

  1. Lines 19-23 and 24: please, explain EAI, ESI P1 and S1.

Response: Many thanks for your constructive comments. We have added the meaning of EAI, ESI, P1 and S1. EAI: emulsifying activity index, ESI: emulsifying stability index, P1: perilla seed oil emulsion with emulsification ratio of 20 % group, S2: soybean oil emulsion with emulsification ratio of 40 %.

  1. Paragraph 2.3. You extract the myofibrillar proteins. Could you please provide an SDS-gel analysis?

Response: Many thanks for your constructive comments. The SDS-PAGE analysis had been performed in our previously published work. The extraction conditions of myofibrillar proteins in the present work were well consistent with the previous methods. Thanks again. 

  1. Paragraph 2.7. Please, list the equations.

Response: Many thanks for your constructive comments. Revised (L137), thanks.

  1. Table 1. What do you mean by 3 and 2 on the left, close to the name of the oil?

Response: Many thanks for your constructive comments. It means the perilla seed oil and soybean oil are both composed of palmitic acid, stearic acid, oleic acid, linoleic acid and linolenic acid.

  1. Figure 1 C. Please, use the complete DLS figure and don't cut the "base". 

Response: Many thanks for your suggestions. Revised in Fig. 1C, thanks.

  1. Figure 4 and 5. Please, adjust the scale. The scale is not the same in the graphs. By consequence, re-write the paragraph 3.5 and 3.6 based on this. 

Response: Many thanks for your constructive comments. We have re-written the paragraph 3.5 and 3.6.

“3.5 Rheological properties

G′ has two maximum values, which are closely related to the types of oil and the composition of fatty acids. Due to the higher proportion of unsaturated fatty acids, the structure transitions were conspicuously observed in the surimi systems using perilla seed oil during heating.

3.6 Protein solubility

The ionic bonds, hydrogen bonds, hydrophobic interactions and disulfide bonds support the three-dimensional network integrity of surimi gel in the different ways. It is thereafter of great significance for underlying thermal stability mechanisms by the insights into the various interactions. determine the technological properties of surimi products [36]. Ionic bonds mainly stabilize the tertiary and quaternary structure of pro-teins, while hydrogen bonds are responsible for secondary structure of proteins. The resultant disulfide bonds from oxidation of sulfhydryl groups support the formation of protein spatial structures. The aggregation protein molecular is mainly mediated by hydrophobic interactions [37,38]. The effects of oil types and emulsification ratio on the protein solubility of surimi gel are shown in Figure 5. The increase of pre-emulsification ratio causes a firstly increased and then decreased change for hydrogen bond, ionic bond and hydrophobic interaction. The peak value is achieved when the pre-emulsification ratio of perilla seed oil and soybean oil is 20 % and 40 %, respective-ly. The similar change to gel properties also suggest that these interaction forces are the inherent contributors. The oppositive results are observed for changes of disulfide bond. The reasons may be the introduction of emulsion systems induces the changes of protein conformation and the corresponding interaction forces [25]. Meanwhile, The disulfide bonds are formed by the oxidation of sulfhydryl groups, which reflect the structural stability of myofibrillar protein. The results reveal that the sulfhydryl groups of myofibrillar protein are not liable to oxidation under the pre-emulsification ratio. The higher structure stability of myofibrillar protein is ascribed to the greater other interactions (hydrogen bonds and hydrophobic interactions), which are also well regarded as the main contributors associating with the structure stability of protein.The introductions of interfacial protein membrane embedding oil droplets contribute to the enhanced hydrogen bonds. The rising pre-emulsification treatments also cause the more expo-sure of hydrophobic groups, and thereby improve hydrophobic interaction. The higher pre-emulsification ratio is achieved by the increasing oil globules in the given protein emulsifier systems. The insufficient amounts of proteins may difficultly meet the requirements of increasing interfacial area, thus leading to the formation of larger oil droplets. The hydrophobic binding sites of oil may be difficultly exposed due to the drop coalescence, which decreased the hydrophobic interaction with protein. These resultant lager oil droplets from over emulsification treatments may fail to stabilize emulsion gel sys-tems due to the weakening hydrophobic interactions, which further affect the struc-ture integrity evidenced by increased disulfide bonds [6] ”.
